# Individual Differences in the Neurocognitive Effect of Movement During Executive Functioning in Children with ADHD: Impact of Subtype, Severity, and Gender

**DOI:** 10.3390/brainsci15060623

**Published:** 2025-06-09

**Authors:** Beverly-Ann Hoy, Maya Feehely, Michelle Bi, Matthew Lam, Androu Abdalmalak, Barbara Fenesi

**Affiliations:** 1Faculty of Education, Western University, London, ON N6G 1G7, Canada; bhoy6@uwo.ca (B.-A.H.); mayafeehely@gmail.com (M.F.); mbi5@uwo.ca (M.B.); mlam324@uwo.ca (M.L.); 2Department of Physiology and Pharmacology, Western University, London, ON N6A 5C1, Canada; aabdalma@uwo.ca

**Keywords:** Attention-Deficit Hyperactivity Disorder (ADHD), physical activity interventions, acute physical activity, fNIRS and physical activity, hypofrontality in ADHD, behavioral ADHD treatment, children with ADHD

## Abstract

**Background/Objectives:** Attention-Deficit Hyperactivity Disorder (ADHD) is an immensely heterogeneous developmental disorder, uniquely impacting each individual. Physical movement is a promising adjunct behavioral treatment that can promote executive functioning in children with ADHD. The current study used neuroimaging and behavioral techniques to investigate the impact of movement during executive functioning on dorsolateral prefrontal cortical (DLPFC) activity and inhibitory control in children with ADHD, with particular focus on key individual difference factors in ADHD, such as subtype, severity, and gender. **Methods**: Twenty-eight children with ADHD completed a Stroop task while remaining stationary (stationary condition) and while desk cycling (movement condition). Simultaneous functional near-infrared spectroscopy (fNIRS) recorded oxygenated and deoxygenated changes in hemoglobin within the left DLPFC. Participants were categorized into ADHD subtype (hyperactive/impulsive, inattention, combined), ADHD severity (low, moderate, high), and gender (male, female). **Results**: Those with the hyperactive and combined ADHD subtypes, those with high ADHD severity, and males with ADHD showed greater DLPFC activation when engaging in movement during executive functioning compared to remaining stationary. In contrast, those with the inattentive ADHD subtype, those with low-to-moderate ADHD severity, and females with ADHD showed greater DLPFC activation when remaining stationary during executive functioning compared to engaging in movement. Inhibitory control improved in the stationary condition for females who were predominantly inattentive. **Conclusions**: This work underscores the importance of considering individual difference factors in ADHD when designing physical activity interventions, as treatment efficacy may vary.

## 1. Introduction

Attention-Deficit Hyperactivity Disorder (ADHD) is a prevalent neurodevelopmental disorder associated with elevated inattention, impulsivity, and hyperactivity in child and adult populations [1,2]. These primary symptoms are related to a compromised executive functioning system, which guides higher-level cognitive functions, such as inhibition, mental flexibility, and working memory [3], all of which are vital for daily function and quality of life. Executive functioning deficits are associated with hypofrontality, a condition characterized by reduced blood flow in the prefrontal cortex (PFC) [4]. Critically, hypofrontality is associated with an impaired hemodynamic response—a slower and less nutrient-dense delivery of blood and oxygen to areas of focal activity [5]. Children with ADHD have been shown to exhibit a diminished hemodynamic response, particularly in the left dorsolateral PFC (DLPFC), which coincides with worse performance on executive functioning tasks (i.e., Stroop, n-back) compared to children without ADHD [4,6,7,8,9,10].

Treatment for ADHD involves a combination of pharmacological and psychosocial intervention approaches [11], with immense variability in therapeutic adherence and efficacy across individuals [12,13]. In general, ADHD around the world is undertreated [14,15,16], and about half of the children with ADHD receive only a single form of treatment, which is often discontinued when parents perceive it as insufficiently addressing academic, social, and family challenges [17]. Despite the complexity and heterogeneity of ADHD [18], intervention research has largely ignored how individual differences, such as ADHD subtype, severity levels, and gender, impact treatment effects [11]. The Aptitude-Treatment Interaction (ATI) framework emphasizes aligning individual characteristics with specific treatment methods to optimize efficacy [19,20]. Despite the ATI framework existing for many decades [20], much of the research on ADHD treatment discounts the potential impact of ADHD heterogeneity on treatment outcomes.

In addition to pharmacological and psychological therapies for ADHD, physical activity has been documented as a beneficial adjunct treatment for children with ADHD [21,22,23]. Critically, physical activity has been shown to target the same catecholaminergic systems (i.e., dopamine, norepinephrine, epinephrine) as stimulant medications [24], promoting more efficient executive functioning and self-regulatory systems. In general, routine physical activity increases cortical thickness and gray matter volume [25,26]; improves energy metabolism and synaptic plasticity [26,27]; stimulates the release of brain-derived neurotrophic factor (BDNF), thereby supporting synaptic and neuronal activity [28,29]; and enhances oxygenation and cortical activation within the prefrontal cortex (PFC) [30]. The neurobiological effect of physical activity is well suited to support the under-aroused PFC characteristic of those with ADHD [2,31,32], helping to promote executive functioning, self-regulation, and mood [33,34,35,36,37]. Physical activity improves core ADHD symptoms (i.e., inattention, hyperactivity, impulsivity), while also addressing associated symptoms such as internalizing and externalizing difficulties, social struggles, and sleep disturbances [21,38]. Educational classrooms that integrate physical activity into instructional time have also been shown to improve several outcomes for children with ADHD, including academic performance, school behavior, classroom engagement, motivation, and on-task behavior [39,40]. In general, physical activity may represent a long-term, protective factor for ADHD through various epigenetic mechanisms, as research indicates higher levels of physical activity in late adolescence are associated with lower ADHD symptoms in early adulthood, even after controlling for genetic and environmental confounding factors [13,41].

Recently, research has shown that engaging in a variety of gross motor movements (e.g., fidgeting, desk-cycling) while completing attention-demanding tasks increases DLPFC blood flow and task performance in children with and without ADHD [3,31,32,42], with a greater magnitude of benefit for those with ADHD [31]. These findings suggest that children with ADHD who engage in more hyperactive movements may be augmenting their attentional capacity. More specifically, hyperactivity may be an adaptive response, helping to mitigate the hypofrontality associated with ADHD, ultimately enhancing executive functioning [31,32]. However, one of the critical gaps in the literature is our understanding of how key individual difference factors in ADHD play a role in the efficacy and suitability of physical activity interventions for those with ADHD. Despite categorical differences in ADHD experience and expression (i.e., inattention, hyperactivity, impulsivity), most intervention studies only include a single ADHD experimental group and do not examine the impact of individual differences on intervention outcomes. Individual differences in ADHD subtype, in symptom severity, and in gender have often been overlooked, leading to standardized treatments that may not be fully effective or tailored to individual needs. Indeed, physical activity interventions for ADHD symptom management may be differentially efficacious depending on such individual difference factors.

### 1.1. Key Individual Difference Factors in ADHD: Subtype, Severity, and Gender

#### 1.1.1. ADHD Subtype

According to the *Diagnostic and Statistical Manual of Mental Disorders* [43], there are three ADHD subtypes: predominantly inattentive (ADHD-I), predominantly hyperactive/impulsive (ADHD-H/I), and combined inattentive/hyperactive (ADHD-C). Behaviorally, the ADHD-I subtype typically exhibits a sluggish cognitive tempo marked by drowsiness, lethargy, and passivity [44], whereas the ADHD-H/I and ADHD-C subtypes are typically more frenetic and hyperactive [45,46]. Neuroimaging studies have identified significant differences in cortical activity between various ADHD subtypes during inhibitory control tasks [47], gray and white matter volume alterations [48], and cortical thickness in several regions [46]. It has even been suggested that the combined and inattentive groups should be considered distinct and unrelated disorders rather than subtypes of the same disorder [44,49,50].

Due to the cognitive, behavioral, and neurobiological differences in ADHD subtypes, variations in the efficacy of physical activity interventions would be expected. This field of inquiry, however, is extremely sparse, with most studies only examining how physical activity interventions impact inattentive or hyperactive symptoms broadly. Recently, studies have shown that physical activity specifically improves inattentive symptoms [51], with large effect sizes of physical activity on inattentive symptoms and medium effect sizes on hyperactive symptoms [52]. In a study directly exploring the association between daily physical activity and affect in individuals with ADHD-I and ADHD-C, researchers found that physical activity, as measured by accelerometers, had the greatest impact on affect in the ADHD-C group [35]. There may be prominent differences in the type of physical activity certain subtypes benefit from the most.

#### 1.1.2. ADHD Severity

ADHD severity has typically been characterized by symptom quantity and persistence, and disorder comorbidities. Psychological and environmental factors that negatively impact executive functioning may contribute to higher ADHD severity. For example, low physical activity levels, overweightness, and obesity are related to low executive functioning, reduced cortical thickness [53,54,55], and ADHD-related cognitive impairments [56,57,58]. Additionally, adverse childhood experiences, parental psychopathology, particularly maternal depression, and socio-economic hardship have been associated with moderate to severe ADHD symptoms in children [59]. In general, engaging in physical activity has been shown to support executive functioning and psycho-emotional wellbeing and reduce symptom severity in children with ADHD [23,34,51,58]. However, to date, there has been no work investigating how physical activity impacts these outcomes across the spectrum of ADHD severity, from low to high. Research in educational interventions suggests that those with greater executive dysfunction may benefit the most from interventions targeting executive functioning, given potentially larger room for improvement [60,61]. Thus, it is possible that those with severe ADHD would benefit the most from physical activity engagement, given the association with greater executive dysfunction.

#### 1.1.3. Gender

In addition to subtype and severity differences, a range of differences in ADHD symptomology exist between males and females. The most widely known gender discrepancy is that males are twice as likely to be diagnosed with ADHD than females [62]. Females are also more likely to receive an ADHD-I diagnosis, whereas males are more likely to be diagnosed with ADHD-C [11,63], with higher rates of hyperactivity and impulsivity [64]. In general, females with ADHD report a greater subjective experience of impairment in several domains, including in home and social life, emotionally, financially, cognitively, and educationally [65,66], despite both genders showing similar levels of impairment on neuropsychological tests [67]. Hyperactive symptoms and conduct problems have been shown to be stronger predictors of clinical diagnosis and prescription of pharmacological treatment than other ADHD symptoms [66,67]. Given the strong presence of internalizing symptoms (depression, anxiety), and fewer hyperactive and externalizing problems, many females are misdiagnosed with exclusively depression and/or anxiety, with the full diagnostic picture being occluded by gender differences in symptom presentation [63,68]. In studies that have examined gender differences in response to pharmacological ADHD treatments (i.e., methylphenidate), no differences have been observed between genders in both pediatric and adult samples [68,69,70]. However, in a review examining psychosocial treatment effects in ADHD, researchers concluded that the literature has largely ignored gender by treatment effects [71]. To date, there has been no research investigating the role of gender in the efficacy of physical activity interventions for ADHD treatment.

### 1.2. Study Objectives

A personalized approach that considers individual differences in ADHD characteristics is crucial for identifying effective treatment outcomes. By utilizing the ATI framework, behavioral, and neuroimaging techniques, the current study investigated the impact of movement during executive functioning on DLPFC activation and inhibitory control in children with ADHD, with a particular focus on individual differences in ADHD subtype, severity, and gender. We hypothesized that movement (compared to remaining stationary) would increase left DLPFC activation and improve inhibitory control (1) for all subtypes, with the greatest magnitude of benefit for the ADHD-H/I group [3,31,32,42,72]; (2) for all severity levels, with the greatest magnitude of benefit for those with severe ADHD, and (3) for both genders, with the greatest magnitude of benefit for males [32,63,72].

## 2. Materials and Methods

### 2.1. Participants

The study sample included 28 children with ADHD. All participants were between 8 and 12 years old (mean age = 9.6, males = 17, females = 11), had an ADHD diagnosis, and were recruited from London, Ontario (Canada). Recruitment flyers were posted at the Western Interdisciplinary Research Building and the Institution’s Mary J. Wright Child & Youth Development Clinic. Flyers were also emailed out to a list of current clients (guardians and children) at the Mary J. Wright Child & Youth Development Clinic, who previously indicated that they would be interested in participating in research studies. Participants were excluded if they were not fully literate or did not speak English, if they had any developmental and neurological exceptionalities above and beyond ADHD, if they were unable to participate in moderate-intensity physical activity, or if they were color-blind.

### 2.2. Measures

#### 2.2.1. Demographic Questionnaire

Demographic information was collected from participants and their guardians (see Table 1). Questions were asked about the guardians’ age, sex, level of education, current employment status, and income. There were also questions concerning the participants’ age, sex, ADHD diagnosis status, other medical diagnoses, prescribed medication, and history of medication.

#### 2.2.2. Vanderbilt ADHD Diagnostic Parent Rating Scale (VADPRS)

The VADPRS was used to augment any missing ADHD-related data (e.g, subtype) and to calculate ADHD severity. The scale consists of the full DSM-V criteria for ADHD, with 18 questions pertaining to inattentive and hyperactive/impulsive symptoms. The questionnaire requires guardians to rate the severity of behaviors on a 4-point scale, ranging from 0 (never) to 3 (very often).

#### 2.2.3. Executive Functioning: Stroop Task

The Stroop task is a widely used measure of inhibitory control in children with ADHD [73,74]. In this study, a keyboard-based version of the Stroop task was utilized (see Figure 1), with specific keys assigned to represent different colors on a 24-inch computer monitor. The designated keys were “D” for red, “F” for green, “J” for blue, and “K” for black, each marked with a corresponding colored sticker for visual reference. During the task, participants were shown color words that either matched the text color (congruent trials) or differed from it (incongruent trials). For instance, in a congruent trial, the word “red” would appear in red ink, requiring the participant to select “D.” In an incongruent trial, the word “red” might appear in blue ink, necessitating a response corresponding to blue (key “J”). Participants were instructed to press on the corresponding key to make a response, and to respond as quickly and accurately as possible. Response times and accuracy (correct vs. incorrect responses) were recorded. The task was developed using Inquisit and consisted of four Stroop task blocks, each lasting 50 s, with 20 s rest intervals between blocks.

#### 2.2.4. Heart Rate

Heart rate was continuously measured by an Apple Watch and the corresponding VeryFit 2.0 app to track desk-cycling intensity. The Apple Watch was secured on the participant’s right wrist and calibrated to ensure consistent and accurate readings prior to beginning the experimental protocol. The “Workout” mode was selected as the physical activity category on the VeryFit app. Heart rate readings were continuously synced from the Apple Watch to the VeryFit app during both conditions. The average heart rate was recorded after each condition.

#### 2.2.5. Left DLPFC Activity Using Functional Near-Infrared Spectroscopy (fNIRS)

The left DLPFC was selected as the region of interest (ROI) based on prior research examining the impact of exercise on Stroop task performance [7,73]. Neural activity within this ROI was measured using functional near-infrared spectroscopy (fNIRS), a non-invasive, portable, and safe neuroimaging technique often recommended for studies involving young participants [74]. The NIRSport system used in this study is particularly suited for experiments involving movement or physical activity [73]. When a brain region becomes active, it consumes oxygen, prompting an increase in blood flow to the area—a process known as neurovascular coupling [74,75]. The influx of oxygen into the tissue is known as oxygenated hemoglobin, and the amount of oxygen absorbed by the tissue is known as deoxygenated hemoglobin. Changes in oxygenated (HbO) and deoxygenated (HbR) hemoglobin concentrations in the blood were measured using fNIRS, as indirect indicators of brain activity [75].

The fNIRS cap (NIRSport2, NIRx Medical Technologies, LLC) used in this study featured an 18-channel system covering the prefrontal cortex, with a sampling rate of 10.2 Hz. Data collection and montage setup were conducted using NIRx’s Aurora software (version 2021.4). Signal quality was assessed using the coefficient of variation (CV) metric and signal level measurements, following NIRx guidelines to ensure CV values remained below 3% and signal levels met or exceeded 3 mV for optimal readings. The montage included eight dual-wavelength sources (760 and 850 nm) and eight detectors, spaced at least 3 cm apart.

During the experiment, the fNIRS cap was securely positioned on the participant’s head, with infrared-emitting optodes placed on the scalp. These optodes emitted light that passed through the skin and skull, where it was absorbed by brain tissue. Nearby detector optodes then measured the amount of light absorption, allowing for the calculation of changes in HbO and HbR levels. These changes were analyzed using the Beer–Lambert law, which accounts for variations in light intensity between emission and detection [74].

### 2.3. Design

The study was within-subjects and counterbalanced, whereby each participant participated in both the movement and stationary conditions in a random order.

#### 2.3.1. Movement Condition

A desk-cycle (3D Innovations Desk-Cycle) was used to facilitate movement while participants completed the desk-based Stroop task [3,25]. To ensure comfort while pedaling, several participant-specific adjustments were made before each experimental session, including modifications to the chair height, the distance from the desk and keyboard, and the positioning of the bike pedals. Participants’ feet were secured in the desk-cycle’s pedals, and the resistance was set to the lowest level. They maintained a consistent, self-selected pedaling pace throughout the experiment. The fNIRS cap remained in place for the entire session. During the movement condition, the average heart rate was 94.28 bpm (SD = 8.64). Figure 2 illustrates the experimental setup.

#### 2.3.2. Stationary Condition

Participants remained stationary with their feet on the floor during the completion of the Stroop task and fNIRS recording. During the stationary condition, the average heart rate was 85.28 (SD = 8.22).

### 2.4. Procedure

The study took place in the Faculty of Education building at Western University. Participants were instructed to refrain from taking medication for 24 h before the study began. However, they had the option to decline this requirement, with their decision being documented. If participants and their guardians agreed to take part in the study, researchers guided them through consent forms and assent forms, as well as the VADPRS and demographic questionnaire. The participant was then taken into an adjacent room and was introduced to the study procedures. They were familiarized with the fNIRS equipment, beginning with a measurement of their head circumference using a fabric measuring tape to determine the appropriate cap size. Participants were given the opportunity to examine the fNIRS cap and touch the infrared optodes to increase their comfort. Researchers addressed any questions or concerns before seating the participants and fitting the cap onto their heads. Participants were asked to remain still for approximately 30 s while researchers optimized the fNIRS channels, ensuring that CV levels were below 3% and signal levels reached at least 3 mV. Once optimization was complete, participants were able to view their brain activity on the display screen and ask any additional questions.

Figure 3 presents a flow diagram of the study procedure. Participants received instructions on the Stroop task and completed several practice Stroop trials before beginning the real task. Each participant completed both the stationary and movement conditions, with the order counterbalanced to ensure random assignment. In the movement condition, participants cycled on a desk-cycle for one minute to establish a stable, comfortable pace before starting the task. In the stationary condition, they began the experiment immediately. The Stroop task consisted of four blocks (50 s each) with 20 s rest intervals between them. At the end of the study, participants and their guardians were debriefed, and researchers addressed any remaining questions.

### 2.5. Determining ADHD Subtype

ADHD subtypes were determined using parental reports on the demographic questionnaire and were based on an official physician assessment. Four parental guardians were unsure of their child’s ADHD subtype, and thus, the VADPRS was used to identify which subtype the child belonged to.

### 2.6. Determining ADHD Severity

Ratings on the VADPRS using the 4-point scale (“never” to “very often”) indicates the severity of each behavior, including behaviors of inattention (i.e., misses details, difficulty sustaining attention, distracted by external stimuli) and hyperactivity/impulsivity (i.e., fidgets, difficulty sitting still, talks excessively). A total score was calculated using the 4-point scale within the inattentive- and hyperactive/impulsive-specific questions (questions 1–18). The total score was then computed into a percentage. Total percentages were sorted from lowest to highest to identify the range of severity and binned into three categories: low severity (31–49%), moderate severity (50–67%), and high severity (68–90%).

### 2.7. Statistical Analysis

#### 2.7.1. fNIRS and Left DLPFC Activity

The fNIRS data were preprocessed and analyzed using Satori (v 2.06). First, the raw data were trimmed by 10 s before and after the last trigger. Raw data were then converted into optical density and corrected for motion artifacts using spike removal and TDDR [25]. Following, the optical density data were converted into concentration changes using the modified Beer–Lambert law and band-pass filtered ([0.01–0.5 Hz]) to remove high-frequency noise. Finally, signals were normalized, and general linear modeling (GLM) analyses were conducted at the individual level. For each participant, the contrast of movement > stationary was examined. The left DLPFC locations on the fNIRS montage correspond to regions F1, F3, and F5 (channels S4-D2, S1-D2, and S1-D1, respectively). Paired samples *t*-tests were conducted to compare HbO and HbR differences in ROI channels between movement and stationary conditions. To be considered a marker of ROI activation, significant activation of at least one channel within the ROI had to be present. Two participants yielded poor fNIRS signal quality and were removed from further analysis.

#### 2.7.2. Executive Functioning

All statistical analyses for executive outcomes were conducted using SPSS 29. Individual difference factors of ADHD subtype, ADHD severity, and gender were analyzed separately. For executive functioning outcomes according to ADHD subtype, four 2 × 3 mixed factorial analysis of variance (ANOVA) were conducted for Congruent and Incongruent Stroop RT and proportion correct outcomes with a within-subject factor of condition (movement vs. stationary) and a between-subject factor of ADHD subtype (inattentive, hyperactive, combined). Exploratory paired samples *t*-tests were performed to compare within-participant differences in outcome measures between movement and stationary conditions separately for each level of subtype. For executive functioning outcomes according to ADHD severity, four 2 × 3 mixed factorial analysis of variance (ANOVA) were conducted for Congruent and Incongruent Stroop RT and proportion correct outcomes with a within-subject factor of condition (movement vs. stationary) and a between-subject factor of ADHD severity (low, moderate, severe). Exploratory paired samples *t*-tests were performed to compare within-participant differences in outcome measures between movement and stationary conditions separately for each level of severity. For executive functioning outcomes according to gender, four 2 × 3 mixed factorial analysis of variance (ANOVA) were conducted for Congruent and Incongruent Stroop RT and proportion correct outcomes with a within-subject factor of condition (movement vs. stationary) and a between-subject factor of gender (male, female). Exploratory paired samples *t*-tests were performed to compare within-participant differences in outcome measures between movement and stationary conditions separately for each gender.

## 3. Results

To check for potential confounding effects of counterbalancing the order of the movement and stationary conditions, four 2 × 2 mixed factorial ANOVA were conducted on Congruent and Incongruent Stroop RT and proportion correct scores, where condition order (stationary-first, movement-first) was included in addition to the factors of the main analysis. Results showed no significant effects of counterbalancing on any EF outcomes (*F*s < 2.76, *p*s > 0.11, all ηp2 < 0.09), indicating that counterbalancing order did not impact the effect of our experimental manipulation. One participant was removed from the gender analyses on the Congruent Stroop proportion correct analysis, and one participant was removed from the subtype analyses on the Congruent Stroop RT analysis, as they were extreme outliers (SPSS step of 1.5 × interquartile range). Heart rate values were significantly higher in the movement condition compared to the stationary condition, t(27) = −5.61, *p* < 0.001, *d* = 1.14.

### 3.1. fNIRS and Left DLPFC Activity

#### 3.1.1. ADHD Subtype

Four of six participants (66%) with the *hyperactive* ADHD subtype showed greater HbO during the movement condition in the left DLPFC (all β*s* −1.00–0.63, *t*s > 2.02, all *p*s < 0.04), whereas only two of six participants (33%) showed greater HbO during the stationary condition compared to the movement condition in the left DLPFC (all β*s* −0.09–1.11, all *t*s < −7.3, all *p*s < 0.001). Similarly, 8 of 15 participants (53%) with the *combined* ADHD subtype showed greater HbO during the movement condition in the left DLPFC (all β*s* −0.96–0.96, all *t*s > 5.40, all *p*s < 0.001), whereas 7 of 15 participants (47%) showed greater HbO during the stationary condition compared to the movement condition in the left DLPFC (all β*s* −1.43–1.51, all *t*s < −3.78, all *p*s < 0.001). In contrast, three of five participants (60%) with the *inattentive* ADHD subtype showed greater HbO during the stationary condition compared to the movement condition in the left DLPFC (all β*s* −0.70–0.86, all *t*s < −2.86 all *p*s < 0.004), whereas two of five participants (40%) showed greater HbO during the movement condition in the left DLPFC (all β*s* −0.57–0.55, all *t*s > 5.45, all *p*s < 0.001). Figure 4, Figure 5 and Figure 6 provide representative images of participants with hyperactive ADHD, inattentive ADHD, and combined ADHD, and their HbO concentration differences during movement vs. stationary conditions while performing the Stroop task. Please refer to Appendix A for a full breakdown of fNIRS *t*-test results.

#### 3.1.2. ADHD Severity

Five of eight participants (63%) with *low severity* ADHD showed greater HbO during the stationary condition compared to the movement condition in the left DLPFC (all β*s* −0.33–1.11, all *t*s < −2.86, all *p*s < 0.004), whereas only three of eight participants (37%) showed greater HbO during the movement condition in the left DLPFC (all β*s* −0.44–0.92, all *t*s > 2.02, all *p*s < 0.04). Five of nine participants (56%) with *moderate severity* ADHD showed greater HbO during the stationary condition compared to the movement condition in the left DLPFC (all β*s* −0.70–1.51, all *t*s < −5.21, all *p*s < 0.001), whereas only four out of nine participants (44%) showed greater HbO during the movement condition in the left DLPFC (all β*s* −1.00–0.26, all *t*s > 5.71, all *p*s < 0.001). In contrast to these patterns, seven of nine participants (78%) with *severe* ADHD showed greater HbO during the movement condition in the left DLPFC (all β*s* −0.96–0.96, all *t*s > 5.40, all *p*s < 0.001), whereas only two of nine participants (22%) showed greater HbO during the stationary condition compared to the movement condition in the left DLPFC (all β*s* −1.44–0.68, all *t*s < −3.78, all *p*s < 0.001). Figure 7, Figure 8 and Figure 9 provide representative images of participants with low, moderate, and severe ADHD and their HbO concentration differences during movement vs. stationary conditions while performing the Stroop task. Please refer to Appendix A for a full breakdown of fNIRS *t*-test results.

#### 3.1.3. Gender

Seven of eleven female ADHD participants (64%) showed greater HbO during the stationary condition compared to the movement condition in the left DLPFC (all β*s* −0.91–1.21, all *t*s < −5.28, all *p*s < 0.001), whereas only four of eleven participants (36%) showed greater HbO during the movement condition in the left DLPFC (all β*s* −0.59–0.96, all *t*s > 2.23, all *p*s < 0.02). In contrast, 10 of 14 male ADHD participants (71%) showed greater HbO during the movement condition in the left DLPFC (all β*s* −1.00–0.93, all *t*s > 5.40, all *p*s < 0.001), whereas only 4 of 14 participants (29%) showed greater HbO during the stationary condition compared to the movement condition in the left DLPFC (all β*s* −1.44–1.51, all *t*s < −3.25, all *p*s < 0.001). Figure 10 and Figure 11 provide representative images of male and female participants and their HbO concentration differences during movement vs. stationary conditions while performing the Stroop task. Please refer to Appendix A for a full breakdown of fNIRS *t*-test results.

### 3.2. Executive Functioning

#### 3.2.1. ADHD Subtype

Table 2 provides EF descriptives for each ADHD subtype. For all EF outcome variables (Congruent Stroop RT, proportion correct; Incongruent Stroop RT, proportion correct), all 2 × 3 mixed factorial ANOVA yielded non-significant main effects of condition (all *F*s < 3.72, all *p*s > 0.07, all ηp2 < 0.08), non-significant main effects of subtype (all *F*s < 1.79, all *p*s > 0.19, all ηp2 < 0.13), and non-significant interactions (all *F*s < 1.26, all *p*s > 0.30, all ηp2 < 0.30). Exploratory paired *t*-test analyses within each level of ADHD subtype yielded similar Stroop outcomes in movement and stationary conditions (all *t*s < 0.98, all *p*s > 0.17, all *d*s = 0.09–0.84).

#### 3.2.2. ADHD Severity

Table 3 provides EF descriptives for each level of ADHD severity. For all EF outcome variables (Congruent Stroop RT, proportion correct; Incongruent Stroop RT, proportion correct), all 2 × 3 mixed factorial ANOVA yielded non-significant main effects of condition (all Fs < 1.62, all ps > 0.22, all ηp2 < 0.06), non-significant main effects of severity (all Fs < 2.38, all ps > 0.11, all ηp2 < 0.16), and non-significant interactions (all Fs < 0.64, all ps > 0.53, all ηp2 < 0.05). Exploratory paired *t*-test analyses within each level of ADHD severity yielded similar Stroop outcomes in movement and stationary conditions (all *t*s < 1.66, all *p*s > 0.07, all *d*s 0.05–0.84).

#### 3.2.3. Gender

Table 4 provides EF descriptives for each gender. For all EF outcome variables (Congruent Stroop RT, proportion correct; Incongruent Stroop RT, proportion correct), all 2 × 3 mixed factorial ANOVA yielded non-significant main effects of condition (all *F*s < 1.94, all *p*s > 0.18, all ηp2 < 0.07), non-significant main effects of gender (all *F*s < 0.74, all *p*s > 0.40, all ηp2 < 0.08), and non-significant interactions (all *F*s < 2.33, all *p*s > 0.14, all ηp2 < 0.03). Exploratory paired *t*-test analyses within the female subgroup yielded significantly greater proportion correct responses on the Congruent portion of the Stroop task in the stationary condition compared to the movement condition (*t*(9) = 2.21, *p* = 0.03, *d* = 1.00), with all other *t*-tests showing similar Stroop outcomes in movement and stationary conditions across both genders (all *t*s < 1.16, all *p*s > 0.16, all *d*s 0.09–0.49).

## 4. Discussion

The current study found differing neural and cognitive benefits of movement during executive functioning depending on individual difference factors of ADHD subtype, ADHD severity, and gender. Specifically, those with the hyperactive and combined ADHD subtypes showed stronger DLPFC activation when engaging in movement during executive functioning compared to remaining stationary; those with high severity ADHD showed stronger DLPFC activation when engaging in movement during executive functioning, whereas those with low-to-moderate severity ADHD showed stronger DLPFC activation when remaining stationary; and lastly, although males showed stronger DLPFC activation when engaging in movement during executive functioning, females showed the opposite, with stronger DLPFC activation during stationary executive functioning. Although most executive functioning outcomes did not parallel DLPFC activation, we did find that inhibitory control was enhanced during the stationary condition for females with ADHD, which matched DLPFC activation patterns. The following sections will discuss the implications of these findings.

ADHD subtype played a significant role in determining the neural response to movement during executive functioning. Most participants with the hyperactive ADHD subtype (78%) showed greater left DLPFC activation when engaging in movement than when remaining stationary. Prior work has shown that allowing children with ADHD to engage in more gross motor movement (desk-cycling, fidgeting) during attention-demanding tasks can serve as a compensatory mechanism that helps upregulate cortical arousal by encouraging more cerebral blood flow and consequently supporting executive functioning [3,31,32,73]. However, the current findings revealed important nuances to these prior conclusions. Only those with the hyperactive subtype (and to a lesser degree, those with the combined subtype) showed DLPFC activation benefits from movement during executive functioning, whereas those with the inattentive ADHD subtype showed the greatest DLPFC activation while remaining stationary (60%). Despite physical activity, in general, offering significant benefits to those with the inattentive subtype [13,51,52], combining physical movement during attention-demanding tasks may overburden an already impaired attentional system among those with inattentive ADHD and function more as a form of multi-tasking or distraction, thereby impairing DLPFC activation. Our results suggest that the benefit of movement during executive functioning may be limited to those with hyperactive symptoms (ADHD-H/I, ADHD-C) due to differences in underlying pathophysiology, whereby bodily movement serves as an augmentation of cortical hypoarousal only in a subgroup of those with ADHD. Treatments directly targeting inattention, such as mindfulness practices [34] during executive functioning, rather than treatments targeting movement-based compensatory mechanisms, may be more effective in improving DLPFC functioning for those with inattentive ADHD. Differences in the neural efficacy of movement in ADHD subtypes may also be connected to previous research demonstrating the benefit of closed motor versus open motor activities. Indeed, closed motor activities that require proprioceptive sensory feedback are beneficial for individuals with hyperactive/impulsive ADHD, whereas open motor activities that require strong visual attention are beneficial for individuals with inattentive ADHD [51]. The desk-cycle is a solid example of a closed motor activity, as it consists of one consistent movement in a stable environment, with the use of more proprioceptive sensory feedback than visual attention.

An important caveat to mention is the disproportionate number of males and females in each subtype; results may have varied with an equal distribution of males and females in each subtype. For example, since females also benefited the most from remaining stationary during executive functioning, and they also constituted the majority of the inattentive subtype, it is unknown whether the mechanism driving the benefits is a gender-mediated mechanism, an inattentive subtype-mediated mechanism, or both. Notwithstanding, the present subtype results are in line with hypotheses and prior research suggesting that hyperactivity is an adaptive tool to regulate hypofrontality [3,32,42], as the only groups who benefited from movement during executive functioning were groups with higher levels of hyperactivity, including those with ADHD-H/I, ADHD-C, and males.

With respect to the impact of individual differences in ADHD severity, we found that only those children with high severity ADHD had greater DLPFC activation when engaging in movement during executive functioning. In contrast, those with low or moderate severity ADHD had greater DLPFC activation when remaining stationary during executive functioning. It is interesting to note that most participants with severe ADHD had the combined subtype, which is consistent with prior research suggesting that ADHD-C exhibits the most impairment [62]. Although psychostimulant medication is typically recommended for children with high severity ADHD (i.e., more ADHD symptoms present), many do not respond to medication, experience disabling side effects, or are left with unaddressed secondary symptoms (e.g., internalizing difficulties, externalizing difficulties, social impairments) [13,76]. Engaging in physical activity in general, or engaging in movement during executive functioning tasks, may help bridge the gap for children with high severity ADHD by improving cortical arousal. Although executive functioning did not improve during the movement condition despite neural benefits for those with severe ADHD, the positive effect on DLPFC activity is nonetheless noteworthy.

Gender also played a significant role in determining the efficacy of movement on DLPFC activity and executive functioning. Females had significantly greater DLPFC activation and inhibitory control performance when remaining stationary compared to when desk-cycling. In contrast, males had significantly greater DLPFC activation when desk-cycling compared to remaining stationary. As previously mentioned, the differential impact of movement during executive functioning may not be due to gender itself, but rather the disproportionate inattentive ADHD subtype in the female population, which does not seem to benefit from movement during executive functioning, as it may overburden an already dysfunctional attentional system. Conversely, males constitute most of the hyperactive ADHD subtype, with hyperactivity (or increased gross motor movement) posited as a beneficial compensatory strategy to promote neural and executive functioning [3,31,32]. These findings point to the ADHD subtype likely playing the largest role in impacting the efficacy of movement-based interventions for children with ADHD, with gender as a related but indirect factor.

Taken all together, findings from our work highlight the importance of understanding how individual factors like ADHD subtype, severity levels, and gender impact the effectiveness of physical activity interventions. ADHD is a highly heterogeneous disorder, and research investigating intervention outcomes based on individual differences in ADHD has been neglected, which may affect the overall treatment and prognosis of those diagnosed with the disorder. The ATI framework provides a purposeful foundation for further research in this area as it emphasizes the need to identify key individual characteristics that may impact the receptivity to different interventions or treatments. The current study demonstrated that ADHD subtype, especially, impacts the neural effects of movement during attention-demanding tasks. Children with high severity ADHD also benefited the most from movement, along with male children. Overall, this study underscores the need for educational professionals and clinicians to carefully consider individual difference factors when prescribing and evaluating ADHD interventions. Given that ADHD treatment should not follow a one-size-fits-all model, a collaborative approach involving both children with ADHD and professionals is essential to help identify a mutual fit between the person and intervention.

### Limitations

The current study has several limitations. First, the ADHD-I and ADHD-H/I groups had relatively small sample sizes due to the higher prevalence of ADHD-C. Second, the gender distribution was uneven across subtypes: the ADHD-I group consisted primarily of females, while the ADHD-C and ADHD-H/I groups were predominantly male. However, these could be considered inherent limitations within ADHD research that investigates individual differences, as these are naturally occurring distribution differences in the ADHD population. However, these limitations do undermine the generalizability of the results, and strong conclusions cannot be drawn; future work should consider recruiting participants a priori who belong to the various ADHD subtypes and consider equalizing gender distribution among the groups. Although this is an ideal suggestion, feasibility with recruitment is always the counterpoint and should be considered as well. Third, participants were not screened for psychiatric disorders beyond the exclusion criteria, and common comorbidities such as oppositional defiant disorder (ODD) and mood and anxiety disorders were not controlled for. Given the high prevalence of comorbid conditions in ADHD, recruiting participants with ADHD-only diagnoses would be exceptionally challenging, if not impossible. Indeed, future work should consider using the Attention Network Task (ANT) as it offers a multidimensional assessment of the complex process of attention by dividing it into three dissociable components: alerting, orienting, and executive control [77,78]. The ANT has been used in previous work with children who have ADHD and holds potential in the context of movement-based interventions in ADHD. Fourth, there was no validated approach to determining severity distribution using the VADPRS; although we used a percentile distribution method to categorize severity, other methods could yield different severity distributions and thus impact the interpretability of the data. Fifth, several factors could have contributed to the DLPFC and executive functioning outcomes not coalescing, such as small sample size, an insufficiently sensitive inhibitory control task, and the fact that neural activity does not always translate into behavioral outcomes. A combination of increased sample size and more diverse and sensitive executive functioning tasks can hopefully better align neural and behavioral outcomes. Sixth, we only used a single task to measure executive functioning (Stroop); using multiple measures to assess inhibitory control could provide a more robust estimate of executive functioning and foster a better match between neural activation and executive functioning. Lastly, although participants refrained from ADHD medication use for 24 h prior to the study, some participants were medicated otherwise, which may have influenced outcomes. However, prior research indicates that medication does not affect the responsiveness of children with ADHD to physical activity interventions.

## 5. Conclusions

This work underscores the importance of considering individual difference factors in ADHD, especially ADHD subtype, when designing physical activity interventions. Although there is strong evidence for the chronic neural and cognitive benefits of engaging in physical activity for children with ADHD, the acute effects of combining physical activity during attention-demanding tasks on neurocognitive function can vary depending on ADHD subtype, severity, and gender.

## Figures and Tables

**Figure 1 brainsci-15-00623-f001:**
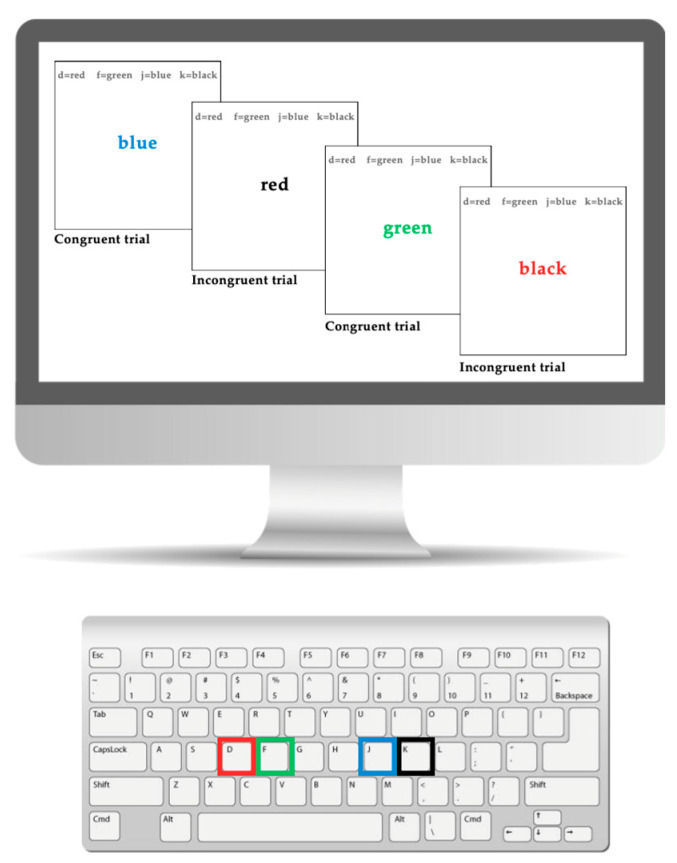
This figure shows the keyboard and computer version of the Stroop task.

**Figure 2 brainsci-15-00623-f002:**
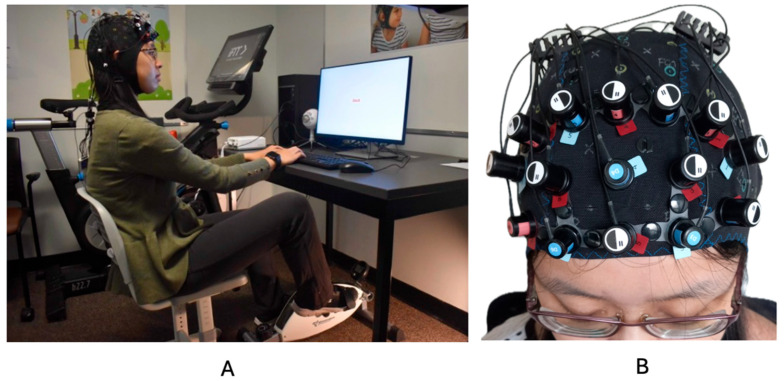
(**A**) depicts a representative participant and the experimental setup. (**B**) depicts the fNIRS cap setup. This image was taken from previously published work Hoy et al., 2024 [31].

**Figure 3 brainsci-15-00623-f003:**
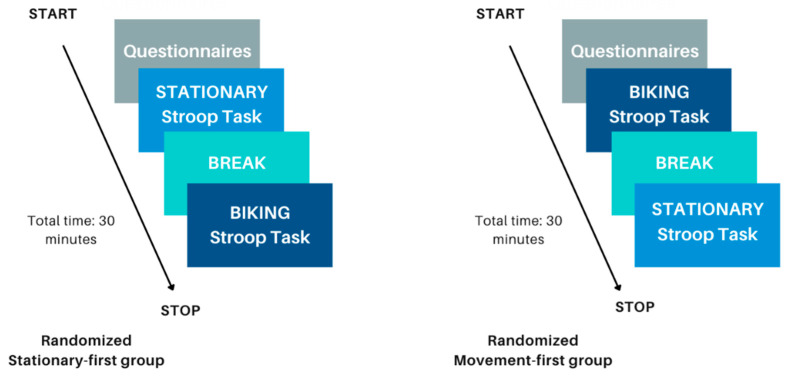
This figure shows the flow of the experimental procedure. The left represents the experimental procedure if a participant was randomized to perform the stationary condition first; the right represents the experimental procedure if a participant was randomized to perform the biking movement condition first.

**Figure 4 brainsci-15-00623-f004:**
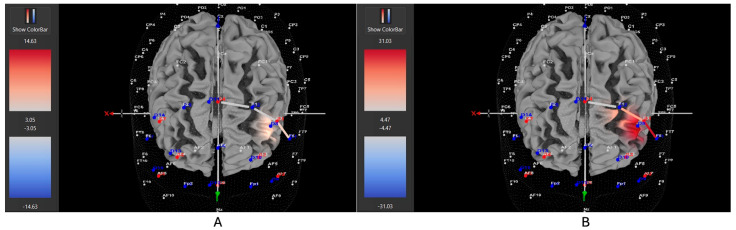
These are representative images of a participant with the inattentive ADHD subtype comparing concentrations of oxygenated hemoglobin in the left DLPFC between the (**A**) movement and (**B**) stationary condition. The red shade indicates greater oxygenated hemoglobin in the regions of interest (i.e., F1, F3, and F5) corresponding to the left DLPFC during the (**B**) stationary condition. The images are masked to only show channels over the ROI. The color scale depicts maximum and minimum activation values for oxygenated hemoglobin (red) and deoxygenated hemoglobin (blue).

**Figure 5 brainsci-15-00623-f005:**
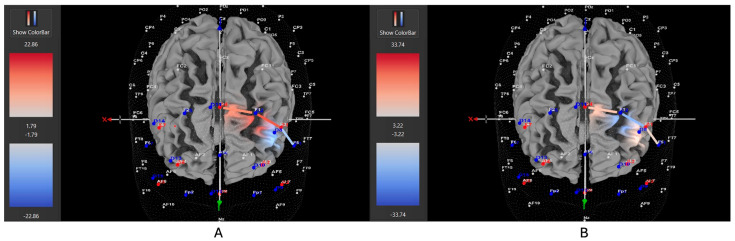
These are representative images of a participant with the hyperactive ADHD subtype comparing concentrations of oxygenated hemoglobin in the left DLPFC between the (**A**) movement and (**B**) stationary condition. The red shade indicates greater oxygenated hemoglobin in the regions of interest (i.e., F1, F3, and F5) corresponding to the left DLPFC during the (**A**) movement condition. The images are masked to only show channels over the ROI. The color scale depicts maximum and minimum activation values for oxygenated hemoglobin (red) and deoxygenated hemoglobin (blue).

**Figure 6 brainsci-15-00623-f006:**
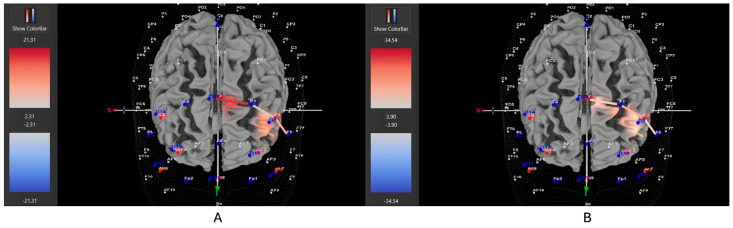
These are representative images of a participant with the combined ADHD subtype comparing concentrations of oxygenated hemoglobin in the left DLPFC between the (**A**) movement and (**B**) stationary condition. The red shade indicates greater oxygenated hemoglobin in the regions of interest (i.e., F1, F3, and F5) corresponding to the left DLPFC during the (**A**) movement condition. The images are masked to only show channels over the ROI. The color scale depicts maximum and minimum activation values for oxygenated hemoglobin (red) and deoxygenated hemoglobin (blue).

**Figure 7 brainsci-15-00623-f007:**
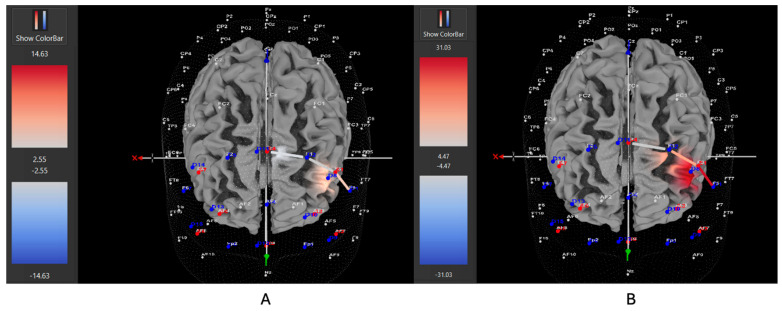
These are representative images of a participant with low severity ADHD comparing concentrations of oxygenated hemoglobin in the left DLPFC between the (**A**) movement and (**B**) stationary condition. The red shade indicates greater oxygenated hemoglobin in the regions of interest (i.e., F1, F3, and F5) corresponding to the left DLPFC during the (**B**) stationary condition. The images are masked to only show channels over the ROI. The color scale depicts maximum and minimum activation values for oxygenated hemoglobin (red) and deoxygenated hemoglobin (blue).

**Figure 8 brainsci-15-00623-f008:**
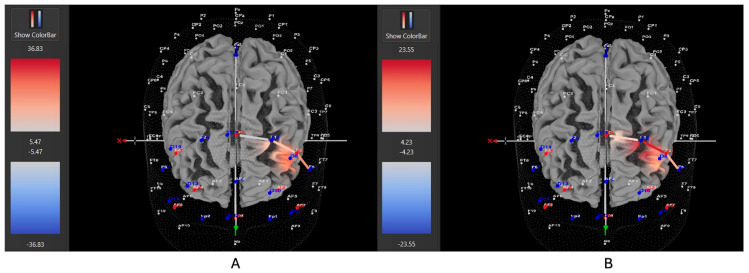
These are representative images of a participant with moderate severity ADHD comparing concentrations of oxygenated hemoglobin in the left DLPFC between the (**A**) movement and (**B**) stationary condition. The red shade indicates greater oxygenated hemoglobin in the regions of interest (i.e., F1, F3, and F5) corresponding to the left DLPFC during the (**B**) stationary condition. The images are masked to only show channels over the ROI. The color scale depicts maximum and minimum activation values for oxygenated hemoglobin (red) and deoxygenated hemoglobin (blue).

**Figure 9 brainsci-15-00623-f009:**
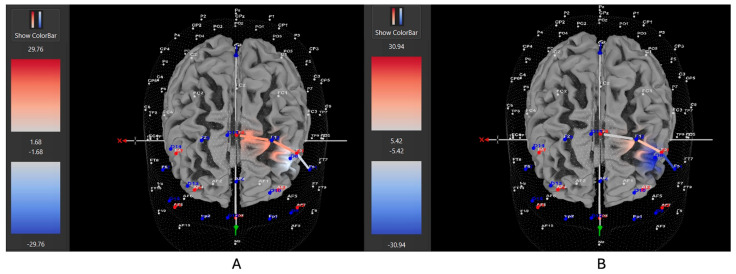
These are representative images of a participant with high severity ADHD comparing concentrations of oxygenated hemoglobin in the left DLPFC between the (**A**) movement and (**B**) stationary condition. The red shade indicates greater oxygenated hemoglobin in the regions of interest (i.e., F1, F3, and F5) corresponding to the left DLPFC during the (**A**) movement condition. The images are masked to only show channels over the ROI. The color scale depicts maximum and minimum activation values for oxygenated hemoglobin (red) and deoxygenated hemoglobin (blue).

**Figure 10 brainsci-15-00623-f010:**
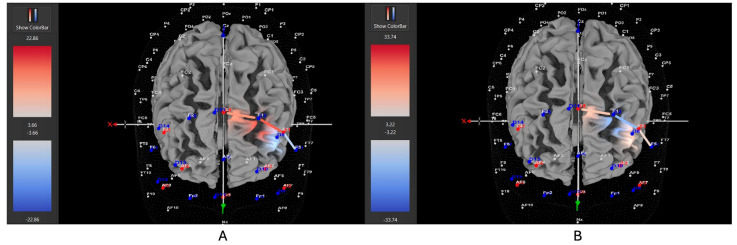
These are representative images of a male participant when comparing concentrations of oxygenated hemoglobin in the left DLPFC between the (**A**) movement and (**B**) stationary condition. The red shade on the right (left hemisphere) indicates greater oxygenated hemoglobin in the regions of interest (i.e., F1, F3, and F5) corresponding to the left DLPFC during the (**A**) movement condition. The images are masked to only show channels over the ROI. The color scale depicts maximum and minimum activation values for oxygenated hemoglobin (red) and deoxygenated hemoglobin (blue).

**Figure 11 brainsci-15-00623-f011:**
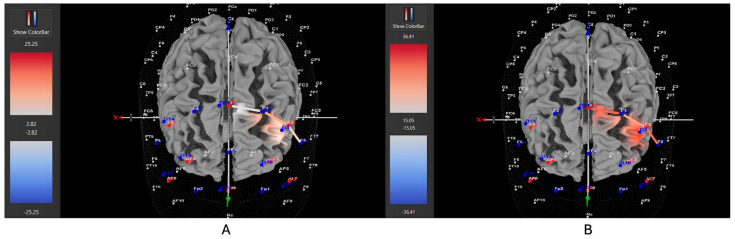
These are representative images of a female participant when comparing concentrations of oxygenated hemoglobin in the left DLPFC between the (**A**) movement and (**B**) stationary condition. The red shade on the right (left hemisphere) indicates greater oxygenated hemoglobin in the regions of interest (i.e., F1, F3, and F5) corresponding to the left DLPFC during the (**B**) stationary condition. The images are masked to only show channels over the ROI. The color scale depicts maximum and minimum activation values for oxygenated hemoglobin (red) and deoxygenated hemoglobin (blue).

**Table 1 brainsci-15-00623-t001:** Demographic information for participants and their guardians.

Characteristic	*n*
Age of Child (years), mean (SD)	9.6 (1.57)
Age of Guardian (years), mean (SD)	40.82 (7.21)
Sex of Participant	
Male	17 (61)
Female	11 (39)
Sex of Guardian	
Male	6 (21)
Female	22 (79)
Guardian’s Education Level (*n*)	
Some high school, no diploma	0 (0)
High school graduate, diploma, or the equivalent	1 (4)
Some college credit, no degree	2 (7)
Trade/technical/vocational training	3 (11)
Associate degree	4 (15)
Bachelor’s degree	9 (33)
Master’s degree	5 (19)
Professional degree	2 (7)
No response	1 (4)
Guardian’s Employment (*n*)	
Employed for wages	19 (68)
Self-employed	5 (18)
Out of work	2 (7)
Homemaker	1 (4)
Student	1 (4)
Household Income	
Prefer not to say	2 (7)
<USD 30,000	1 (4)
USD 30,000–40,000	0 (0)
USD 40,000–50,000	1 (4)
USD 50,000–60,000	3 (11)
USD 60,000–70,000	3 (11)
USD 70,000–80,000	0 (0)
USD 80,000–90,000	2 (7)
USD 90,000–100,000	3 (11)
>USD 100,000	12 (44)
Age of ADHD diagnosis (*n*)	
Unsure	3 (11)
3–5	4 (14)
6–8	15 (54)
9–12	6 (21)
ADHD Subtype (*n*)	
Predominantly inattentive	5 (18)
Predominantly hyperactive/impulsive	6 (11)
Combined subtype	17 (61)
ADHD Severity (*n*)	
Low	10 (36)
Moderate	9 (32)
High	9 (32)
Currently Taking Medication	
No response	4 (14)
Yes	17 (61)
No	7 (25)
Other Diagnosis Present	
Yes	9 (32)
No	19 (68)
Medicated for Another Diagnosis	
Yes	6 (21)
No	22 (79)

**Table 2 brainsci-15-00623-t002:** Descriptive statistics for Stroop task across ADHD subtype and condition.

	Inattentive (*n* = 5)		Hyperactive/Impulsive (*n* = 6)		Combined (*n* = 17)	
	Stationary M (SD)	Movement M (SD)	Stationary M (SD)	Movement M (SD)	Stationary M (SD)	Movement M (SD)
**Congruent RT**	1077 (577)	931 (306)	1621 (1663)	1004 (227)	1117 (547)	1010 (366)
**Incongruent RT**	916 (290)	1011 (117)	1173 (608)	1256 (421)	1251 (464)	1092 (494)
**Congruent PC**	0.95 (0.11)	0.84 (0.15)	0.91 (0.21)	0.95 (0.11)	0.98 (0.06)	0.96 (0.12)
**Incongruent PC**	0.82 (0.18)	0.91 (0.13)	0.82 (0.21)	0.96 (0.11)	0.93 (0.11)	0.94 (0.12)

Note. PC = proportion correct; M = mean; SD = standard deviation.

**Table 3 brainsci-15-00623-t003:** Descriptive statistics for Stroop task across ADHD severity and condition.

	Low Severity		Moderate Severity		High Severity	
	Stationary M (SD) (*n* = 10)	Movement M (SD) (*n* = 10)	Stationary M (SD) (*n* = 9)	Movement M (SD) (*n* = 9)	Stationary M (SD) (*n* = 9)	Movement M (SD) (*n* = 9)
**Congruent RT**	1272 (1325)	1010 (275)	1145 (480)	829 (224)	1230 (659)	1144 (418)
**Incongruent RT**	1106 (521)	1011 (239)	1143 (379)	1028 (334)	1283 (538)	1310 (622)
**Congruent PC**	0.95 (0.16)	0.90 (0.17)	0.97 (0.08)	0.92 (0.12)	0.97 (0.09)	1.00 (0.00)
**Incongruent PC**	0.82 (0.16)	0.90 (0.13)	0.92 (0.17)	0.97 (0.08)	0.93 (0.11)	0.94 (0.12)

Note. PC = proportion correct; M = mean; SD = standard deviation.

**Table 4 brainsci-15-00623-t004:** Descriptive statistics for Stroop task across gender and condition.

	Males		Females	
	Stationary M(SD) (*n* = 17)	Movement M(SD) (*n* = 17)	Stationary M(SD) (*n* = 11)	Movement M(SD) (*n* = 11)
**Congruent RT**	1292 (1025)	1013 (380)	1105 (643)	967 (219)
**Incongruent RT**	1213 (472)	1151 (508)	1117 (494)	1054 (289)
**Congruent PC**	0.95 (0.13)	0.97 (0.09)	0.98 (0.08)	0.89 (0.17)
**Incongruent PC**	0.89 (0.16)	0.94 (0.11)	0.87 (0.16)	0.93 (0.12)

Note. PC = proportion correct; M = mean; SD = standard deviation.

## Data Availability

The data presented in this study are available upon request from the corresponding author. The data are not publicly available due to ethical restrictions.

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
