# Peer review of "Individual Differences in the Neurocognitive Effect of Movement During Executive Functioning in Children with ADHD: Impact of Subtype, Severity, and Gender"

_brainsci, 2025, doi:10.3390/brainsci15060623_

Round 1

Reviewer 1 Report

Comments and Suggestions for Authors

Dear Editor,

Thank you for granting me the opportunity to review this manuscript. However, there are several major issues that should be addressed before it can be considered for publication:

  1.  The introduction section is overly long. I strongly recommend that the authors rewrite this section to enhance the readability and logical flow of the manuscript. Many sentences currently included in the introduction would be more appropriately placed in the discussion section.

  2. I was unable to identify the novelty of this manuscript. Previous studies have already demonstrated that fNIRS can effectively detect alterations in brain function associated with ADHD, particularly in the prefrontal cortex (PFC), which is crucial for executive functions such as response inhibition and working memory. For example, studies have shown that children with ADHD exhibit reduced activation in the right prefrontal cortex during executive function tasks, such as the Go/No-Go task, compared to typically developing children (PMID: 37056303; PMID: 36937678). To improve the readability and logical structure of the manuscript, the authors should discuss and compare their findings with previous studies, and clearly highlight the novelty of their work.

Reviewer 2 Report

Comments and Suggestions for Authors

The paper entitlted "Individual differences in the neurocognitive effect of movement during executive functioning in children with ADHD: impact of subtype, severity and gender" is a well-designed, innovative study that explores highly relevant dimensions of ADHD research. With increased statistical power and methodological refinements, it could significantly influence personalized, movement-based interventions in neurodevelopmental disorders. Using fNIRS during a Stroop task under stationary and movement (desk cycling) conditions, the study finds differential neurocognitive responses and behavioral outcomes across subgroups of ADHD patients. This is  a timely research aligning with trends toward tailored non-pharmacological ADHD interventions.

A. GENERAL COMMENTS:
================

(1) The main limitation lies in the sample size (e.g., n=5 and n=6 for inattentive and hyperactive subtypes, respectively) with lack of power analysis and the absence of controls matching age and gender of the patients. Hence, the authors must clearly mention this limitation and consider the speculative nature of some of their conclusions.

(2) Among the confounding factors, the uneven gender distribution across subtypes is likely to play a crucial role. The authors report a strong gender effect in the comparative fNIRS analysis of HbO in the left DLPFC with respect to Stationary vs. Movement conditions. The Discussion must be improved with this matter.

(3) Subtype and severity classification: the Authors should mention with more details what are the diagnostic procedure followed to classify ADHD subtypes because it seems that cmorbid conditions were not systematically screened or analyzed.

(4) Results section:  No Cohen’s d, eta squared, or other effect size metrics are reported!  Including effect sizes is crucial, especially for borderline or non-significant results. Please report effect sizes throughout the Results Section.

(5) A paragraph could be added to the Discussion (e.g., after line 688) for a broader exploration of executive functioning beyond inhibitory control as measured by the Stroop task. Specifically, the authors should consider discussing the utility of the Attention Network Task (ANT) in future work. The ANT offers a multidimensional assessment of attention by parsing it into three dissociable components: alerting, orienting, and executive control, which aligns well with the study's focus on individual differences in ADHD subtype, severity, and gender. To this aim, selected references [R1-R4] should be considered to strengthen the rationale for multi-component attention measures, particularly in the context of movement-based or neuromodulatory interventions in ADHD.

B. SPECIFIC COMMENTS:
=================

(1) Page 8, L 285–292, Page 9 L 333, 341-342: are the heart rate values (i.e., 94.28 bpm , 85.28) significantly different? how the heart rates can be compared against established pediatric heart rate values? against values measured in other studies?

(2) Page 11, L 386–390: "Total percentages were sorted from lowest to highest… low severity (31%-49%), moderate severity (50-67%), and high severity (68%-90%)... "
 Please explain how these specific percentage thresholds were chosen. Were they based on a validated scale or percentile distributions? This decision can substantially impact interpretability of subgroup results.

(3) Page 17, L 571–575: The authors report non-significant results without discussing on whether this might be due to underpowered subgroups, task insensitivity, or suggest that neural activation patterns do not always translate into behavioral outcomes.

(4) Figures 4-11: embedded legend or caption clarification should be used in order to ease the reading of visual data.

Citations:

[R1] Konrad K, Neufang S, Hanisch C, Fink GR, Herpertz-Dahlmann B. (2006) Dysfunctional attentional networks in children with attention deficit/hyperactivity disorder: evidence from an event-related functional magnetic resonance imaging study. Biol Psychiatry. 59(7):643-51. doi: 10.1016/j.biopsych.2005.08.013.

[R2] Adólfsdóttir S, Sørensen L, Lundervold AJ. (2008) The attention network test: a characteristic pattern of deficits in children with ADHD. Behav Brain Funct. 4:9. doi: 10.1186/1744-9081-4-9.

Round 2

Reviewer 1 Report

Comments and Suggestions for Authors

i have no more question at this stage